# GeoFunFlow: Geometric Function Flow Matching for Inverse Operator Learning over Complex Geometries

## Abstract

Inverse problems governed by partial differential equations (PDEs) are crucial in science and engineering. They are particularly challenging due to ill-posedness, data sparsity, and the added complexity of irregular geometries. Classical PDE-constrained optimization methods are computationally expensive, especially when repeated posterior sampling is required. Learning-based approaches improve efficiency and scalability, yet most are designed for regular domains or focus primarily on forward modeling. To fill this gap, we introduce *GeoFunFlow*, a geometric diffusion framework for inverse problems on complex geometries. GeoFunFlow combines a novel geometric functional autoencoder (GeoFAE) and a latent diffusion model trained via rectified flow. GeoFAE employs a Perceiver module to process unstructured meshes of varying sizes and produces continuous reconstructions of solution fields, while the diffusion model enables posterior sampling from sparse and noisy data. Across five standard benchmarks, GeoFunFlow achieves state-of-the-art reconstruction accuracy over complex geometries, provides calibrated uncertainty quantification, and delivers efficient inference compared to operator-learning and diffusion baselines.

## 1 Introduction

Inverse problems form a broad and foundational class of tasks across science and engineering, where the goal is to recover unknown physical states from indirect or incomplete observations. They arise in diverse applications: medical imaging (e.g., reconstructing tissue properties in MRI or CT (Natterer & Wübbeling, 2001)), geophysics (e.g., inferring subsurface structures from seismic data (Tarantola, 2005)), and fluid dynamics (e.g., recovering flow fields from limited sensor measurements (Cotter et al., 2009)), among many others. These problems are inherently ill-posed, and the difficulty is further compounded when the underlying domain is irregular, as the domain geometry both dictates PDE boundary conditions and introduces substantial variability that must be accurately captured for reliable inference. Classical PDE-constrained optimization methods can yield accurate reconstructions but are computationally prohibitive at scale, particularly when repeated posterior sampling is required.

Operator learning has recently emerged as a powerful paradigm for approximating PDE solution maps, offering rapid evaluation and strong generalization across varying inputs. Foundational works such as the Deep Operator Network (DeepONet) (Lu et al., 2021) and the Fourier Neural Operator (FNO) (Li et al., 2021), along with their extensions, have demonstrated impressive performance across diverse scientific domains, including weather forecasting (Pathak et al., 2022), micromechanics (Wang et al., 2024), astrophysics (Mao et al., 2023), and geosciences (Zhu et al., 2023). Despite these advances, most approaches remain tailored to structured, regular domains where spectral or convolutional representations are naturally applicable. Recent efforts have sought to extend operator learning to complex and irregular geometries (Li et al., 2023b; Wu et al., 2024), but these developments primarily target forward tasks, and inverse problems on irregular domains remain largely underexplored.

In parallel, diffusion-based generative models have emerged as a complementary approach for scientific machine learning. Building on their success in vision and language (Ho et al., 2020; Song et al., 2020), diffusion models have enabled breakthroughs in biological applications such as protein generation (Watson et al., 2023), materials discovery (Zeni et al., 2025), and drug design (Schneuing

et al., 2024). Recent advances also demonstrate their promise in physical simulations, including turbulence modeling (Du et al., 2024), weather forecasting (Li et al., 2024a), metamaterials design (Bastek & Kochmann, 2023), and high-dimensional dynamics prediction (Li et al., 2024b; Gao et al., 2024; Shysheya et al., 2024). However, within PDE neural surrogates, most existing efforts remain limited to regular domains, and consequently extending diffusion-based methods to families of complex geometries remains an open challenge.

In this work, we address this challenge by focusing on inverse problems of reconstructing physical fields governed by PDEs over complex geometries. We introduce GeoFunFlow, a geometry-aware diffusion framework that bridges operator learning and generative modeling. Our main contributions are summarized as follows:

- We propose a geometric functional autoencoder (GeoFAE) that integrates a Perceiver module to handle unstructured meshes of varying sizes, and enables continuous reconstruction of solution fields.
- We seamlessly couple GeoFAE with a latent diffusion model, enabling posterior sampling conditioned on sparse and noisy sensor data.
- We provide theoretical guarantees that, given well-trained GeoFAE and diffusion models, our framework can yield accurate posterior approximations.
- We demonstrate state-of-the-art accuracy and efficient inference on benchmarks involving complex geometries, significantly outperforming representative baselines.

## 2 RELATED WORK

Our approach is closely related to two active directions in scientific machine learning: operator learning for approximating PDE solution operators, and diffusion-based generative models for physical simulations and inverse problems.

**Operator learning.**   Operator learning has emerged as a powerful framework for approximating mappings between infinite-dimensional function spaces. Pioneer architectures such as DeepONet (Lu et al., 2021) and Fourier Neural Operators (FNO) (Li et al., 2021) inspired a series of variants, including UNO (Rahman et al., 2022), WNO (Tripura & Chakraborty, 2022), LNO (Cao et al., 2024), CNO (Raonic et al., 2023), and SNO (Fanaskov & Oseledets, 2023). More recent transformer-based approaches, such as OFormer (Li et al., 2022), FactFormer (Li et al., 2023a), MPP (McCabe et al., 2023), DPOT (Hao et al., 2024), Poseidon (Herde et al., 2024), and CViT (Wang et al., 2025b), leverage attention mechanisms to improve scalability and expressivity. Beyond regular grids, significant progress has been made on complex geometries and irregular meshes, with methods such as Geo-FNO (Li et al., 2023b), GINO (Li et al., 2023c), CORAL (Serrano et al., 2023), AROMA (Serrano et al., 2024), Position-induced Transformer (Chen & Wu, 2024), Transolver (Wu et al., 2024), and PCNO (Zeng et al., 2025), which incorporate geometric priors to extend operator learning to irregular domains.

**Diffusion models for operator learning.**   Diffusion-based generative models have recently been combined with operator learning to address forward and inverse PDE problems. Methods such as DiffFNO (Liu & Tang, 2025) and the Wavelet Diffusion Neural Operator (Hu et al., 2024) integrate diffusion processes with neural operator architectures to capture spatiotemporal dynamics directly from data. Other approaches extend generative models to function spaces for scientific data, including Denoising Diffusion Operators (Lim et al., 2023), and FunDiff (Wang et al., 2025a). In parallel, several works incorporate physics-based inductive biases, such as DiffusionPDE (Huang et al., 2024), physics-informed diffusion models (Bastek et al., 2024), and physics-constrained flow matching (Utkarsh et al., 2025), which enforce PDE constraints during training or sampling to improve fidelity and consistency. However, most of these methods are limited to data defined on uniform grids, restricting their applicability to more general geometric settings.

## 3 PRELIMINARIES

We briefly review the key concepts underlying our approach, namely operator learning for PDE solution maps and rectified flow for generative posterior sampling.

**Operator Learning.** Let $\Omega \subset \mathbb{R}^d$ be a bounded domain and define the input and output function spaces $\mathcal{U} \subset (L^p(\Omega))$ and $\mathcal{V} \subset (L^q(\Omega))$. A target operator $\mathcal{G}^\star : \mathcal{U} \to \mathcal{V}$ maps inputs $a \in \mathcal{U}$ to outputs $u = \mathcal{G}^\star(a) \in \mathcal{V}$. A canonical example is a PDE solution operator, where coefficients or forcing $a = (\kappa, f, g)$ determine $u$ via

$$\mathcal{L}_\kappa u = f \quad \text{in } \Omega, \quad \mathcal{B}u = g \quad \text{on } \partial\Omega.$$

In practice, inputs are drawn from a distribution $\mu$ on $\mathcal{U}$, and outputs must be discretized for training. Here the training data are $(a_i, y_i)_{i=1}^n$ with

$$u_i = \mathcal{G}^\star(a_i), \quad y_i = \mathcal{S}(u_i),$$

where $\mathcal{S} : \mathcal{V} \to \mathbb{R}^{\text{obs}}$ is a sampling operator (e.g., point sensors, projections). Neural operators $(\mathcal{G}_\theta)_{\theta \in \Theta} \subset \mathcal{M}(\mathcal{U}, \mathcal{V})$ are trained via

$$\min_{\theta \in \Theta} \widehat{R}_n(\theta) = \frac{1}{n} \sum_{i=1}^n \ell(\mathcal{S}\mathcal{G}_\theta(a_i), y_i),$$

with $\ell(v, w) = \|v - w\|_{l^2}^2$.

**Rectified Flow (RF).** Rectified Flow (RF) replaces the stochastic dynamics of diffusion models with a deterministic ODE that maps noise to data along straighter trajectories. This yields faster and more stable generation. In this work, we use RF for conditional generation, focusing on posterior estimation $p(x \mid c)$, where $x$ denotes the unknown data and $c$ the observed condition.

Specifically, RF aims to learn a time-dependent velocity field $v_\theta(x_t, t)$ via the ODE

$$\frac{dx_t}{dt} = v_\theta(x_t, t), \quad t \in [0, 1], \tag{1}$$

with initial condition $x_0 \sim \pi_0$ (e.g., Gaussian noise). The flow is trained by minimizing the objective

$$\mathcal{L}_{\text{RF}}(\theta) = \mathbb{E}_{t \sim U[0,1], (x_0, x_1) \sim \pi} \|v_\theta(x_t, t) - (x_1 - x_0)\|^2, \tag{2}$$

where $x_t = (1-t)x_0 + tx_1$ interpolates between $x_0$ and data $x_1 \sim \pi_1$. This yields simpler transport paths compared to score-based diffusion models.

For conditional generation, we extend to $v_\theta(x_t, t, y)$ with objective

$$\mathcal{L}_{\text{CRF}}(\theta) = \mathbb{E}_{t \sim U[0,1], (x_0, x_1, c) \sim \pi} \|v_\theta(x_t, t, c) - (x_1 - x_0)\|^2. \tag{3}$$

Given a trained velocity field, posterior sampling is performed by solving

$$\frac{dx_t}{dt} = v_\theta(x_t, t, c), \quad x_0 \sim \pi_0, \tag{4}$$

and returning the terminal state $x_1 \sim p_\theta(x \mid c)$. This enables efficient posterior sampling without explicit likelihood evaluation or Markov Chain Monte Carlo (MCMC).

## 4 PROBLEM SETUP

We study inverse problems governed by PDEs on irregular geometries. Let $\Omega \subset \mathbb{R}^d$ denote a bounded domain with boundary $\partial\Omega$, where physical states $u : \Omega \to \mathbb{R}^n$ are determined by geometry-dependent dynamics, initial and boundary conditions. The task is to reconstruct $u$ from sparse and noisy sensor measurements, assuming $\Omega$ is given. This setup is canonical in scientific computing, arising in flow reconstruction around aerodynamic bodies, subsurface imaging in geophysics, and biomedical field estimation. The main challenge is achieving reliable inference under limited data and varying geometries.

Formally, across a family of geometries $\{\Omega\}_{\Omega \in \Lambda}$, we consider fields

$$u \in \mathcal{U}(\Omega) := H^s(\Omega; \mathbb{R}^p), \quad s \geq 1,$$

constrained by a geometry-dependent forward operator $\mathcal{F}_\Omega$ (e.g., PDE residuals, boundary conditions). For each $\Omega$, we observe noisy pointwise samples at sensor locations $X = \{x_j\}_{j=1}^m \subset \Omega$:

$$\mathbf{y} = \mathcal{S}_{\Omega, X}[u] + \epsilon, \quad \epsilon \sim \mathcal{N}(0, \Sigma),$$

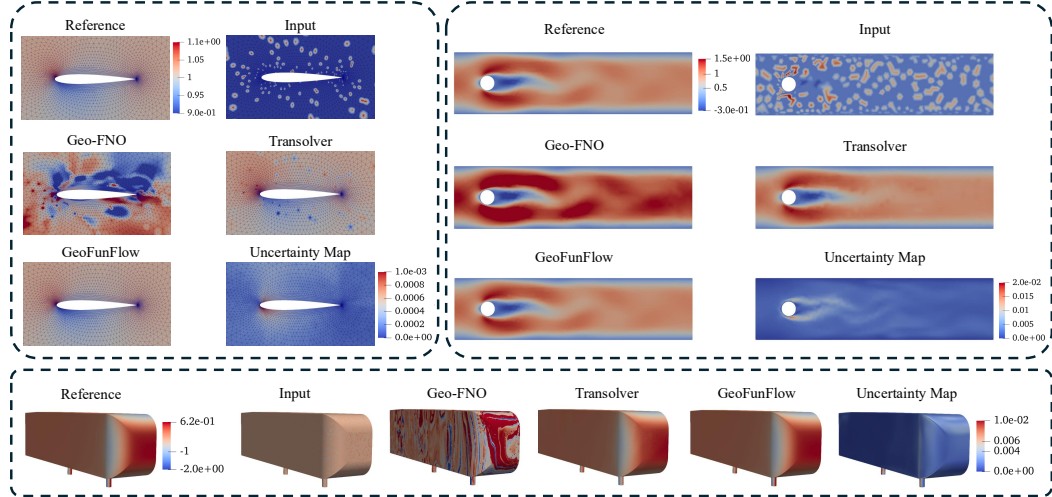

Figure 1: Reconstruction of the target field from noisy (10%) and sparse observations (5%) on the *Airfoil* (left), *Cylinder* (right), and *Ahmed body* (bottom) datasets. For each dataset, we show one representative sample comparing our approach against several competitive baselines.

where $\mathcal{S}_{\Omega,X}$ is the linear observation operator (e.g., pointwise sampling).

Let $\mathcal{P}_2(\mathcal{U}(\Omega))$ denote the space of probability measures over $\mathcal{U}(\Omega)$ with finite second moment, and let $W_2^\Omega$ denote the 2-Wasserstein distance on $\mathcal{P}_2(\mathcal{U}(\Omega))$. A conditioning instance is defined as

$$\mathbf{c} = (\Omega, X, \boldsymbol{y}) \in \mathcal{C} \subset \Lambda \times \mathcal{X}_{\text{sens}} \times \mathbb{R}^{\text{obs}}, \tag{5}$$

where $\mathcal{X}_{\text{sens}}$ encodes admissible sensor configurations. Our goal is to learn a conditional probabilistic operator

$$\mathcal{G}^\star : \mathcal{Y} \to \bigsqcup_{\Omega \in \Lambda} \mathcal{P}_2(\mathcal{U}(\Omega)), \qquad \mathcal{G}^\star(\mathbf{c}) = p^*(\cdot \mid \mathbf{c}), \tag{6}$$

that maps $(\Omega, X, \boldsymbol{y})$ to the posterior distribution of fields on the corresponding geometry $\Omega$, while generalizing across unseen domains.

This formulation highlights three challenges: (i) inferring continuous fields from sparse and noisy sensor data, (ii) extending generative models to tackle varying discretizations across diverse geometries, and (iii) capturing posterior uncertainty rather than deterministic reconstructions. In the next section, we introduce GeoFunFlow, which addresses these challenges by combining a geometric autoencoder with rectified flow–based posterior sampling.

## 5 METHODS

In this section, we present GeoFunFlow to solve the above formulated problem. As shown in Figure 2, our method combines (i) a novel geometric function autoencoder (GeoFAE), which learns the latent representations of fields while enabling continuous reconstruction, and (ii) a conditional Diffusion Transformer (DiT), which models posterior distributions in the latent space via rectified flows.

This decoupled design is motivated by practical considerations. Reconstructing solution fields requires continuous neural representations that generalize across arbitrary sensor locations, while applying diffusion models directly to fields on irregular meshes is computationally prohibitive and often unstable due to large-scale point clouds and varying discretizations. By encoding fields into a compact latent space with GeoFAE, we obtain geometry-aware representations that make posterior inference with diffusion both scalable and stable.

We first introduce our core component, GeoFAE. To instantiate the abstract setup, we discretize each geometry $\Omega$ into a point cloud $V_\Omega \subset \Omega$, where $V_\Omega$ are sampled nodes of the domain. A subset $X \subset V_\Omega$ corresponds to sensor locations with available observations.

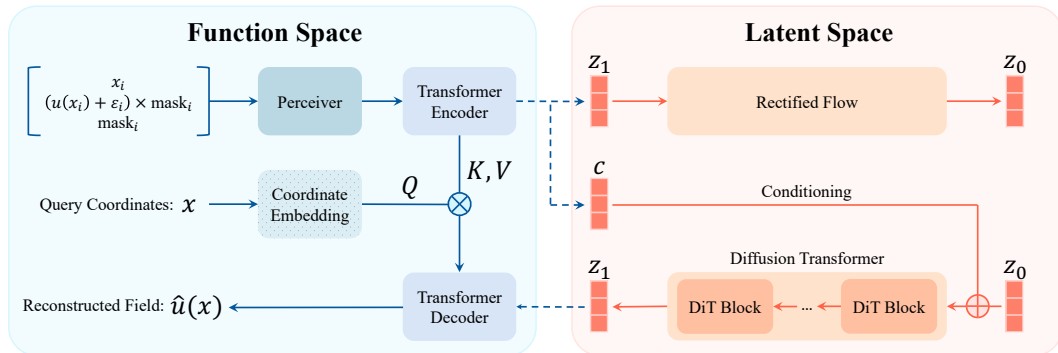

Figure 2: Overall pipeline of GeoFunFlow, which combines a geometric function autoencoder (GeoFAE) with a latent diffusion model. The inputs are (masked noisy) sensor measurements of the target field over a geometry. The GeoFAE encoder maps these inputs into a compact latent representation, while the decoder enables continuous reconstruction at arbitrary query coordinates. In the latent space, a conditional Diffusion Transformer (DiT) performs rectified flow dynamics guided by encoded observations, aligning with the true posterior. At inference, latent samples are decoded back to the physical domain, producing posterior field samples that generalize across diverse geometries and discretizations.

For each node $x \in V_\Omega$, we then define the mask function $M_X(x) = \mathbb{1}_{\{x \in X\}}$, which indicates whether the node is observed. The corresponding observation feature is then $u(x_i) \cdot M_X(x_i)$ equal to the measured value at sensor nodes and zero otherwise. Therefore, a conditioning instance is represented as

$$\mathbf{c} = \big( x_i, M_X(x_i), u(x_i) \cdot M_X(x_i) \big)_{i=1}^{m},$$

that is, the geometry point cloud along with the node-wise mask and observation features.

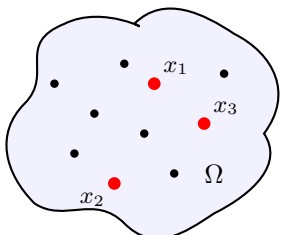

The goal of GeoFAE is to reconstruct the target field $u$ over a known domain $\Omega$ from arbitrary discretizations and sparse observations. The model adopts an autoencoder architecture: the encoder maps the conditioning data to a compact latent representation, and the decoder combines this latent code with query coordinates to continuously recover field values across the domain.

Figure 3: An irregular domain $\Omega$, discretized into a point cloud $V_\Omega$ (black). A subset $X \subset V_\Omega$ (red) represents sensor locations with available observations.

**Encoder.** The encoder $\mathcal{E}_\theta$ employs a Perceiver module (Jaegle et al., 2021) to process varying discretizations. Geometry nodes $\{x_i\}_{i=1}^{m} \in \mathbb{R}^{m \times d}$ are embedded using random Fourier features (Tancik et al., 2020), while the node features $\big( M_X(x_i), u(x_i) \cdot M_X(x_i) \big)_{i=1}^{m}$ are encoded through MLPs. The resulting embeddings $\mathbf{z} \in \mathbb{R}^{m \times D}$ are concatenated and then passed to the Perceiver block.

The Perceiver block introduces trainable latent queries $\mathbf{z}_q \in \mathbb{R}^{P \times D}$, which interact with the input features $\mathbf{z}$ through cross-attention:

$$\mathbf{z}' = \mathbf{z}_q + \mathrm{MHA}\left( \mathrm{LN}\left( \mathbf{z}_q \right), \mathrm{LN}(\mathbf{z}), \mathrm{LN}(\mathbf{z}) \right), \quad \mathbf{z}_{\mathrm{agg}} = \mathbf{z}' + \mathrm{MLP}\left( \mathrm{LN}\left( \mathbf{z}' \right) \right), \quad (7)$$

where MHA, LN, and MLP denote multi-head attention, layer normalization, and a feedforward network, respectively. Here, $P$ is a user-specified parameter that determines the number of queries, and $D$ denotes the latent dimension.

The aggregated tokens $\mathbf{z}_{\mathrm{agg}}$ are processed by $L$ pre-norm Transformer blocks (Vaswani et al., 2017; Xiong et al., 2020):

$$\mathbf{z}_0 = \mathrm{LN}(\mathbf{z}_{\mathrm{agg}}),$$
$$\mathbf{z}'_\ell = \mathrm{MSA}\big( \mathrm{LN}(\mathbf{z}_{\ell-1}) \big) + \mathbf{z}_{\ell-1}, \quad \ell = 1, \dots, L,$$
$$\mathbf{z}_\ell = \mathrm{MLP}\big( \mathrm{LN}(\mathbf{z}'_\ell) \big) + \mathbf{z}'_\ell, \quad \ell = 1, \dots, L,$$

with MSA denoting multi-head self-attention.

It is worth noting that the use of Perceiver compresses the point cloud into a fixed set of latent queries through cross-attention, which reduces the cost of subsequent self-attention and enables the encoder to process meshes of varying sizes while maintaining global context.

**Decoder.** The decoder $\mathcal{D}_\phi$ builds on CViT (Wang et al., 2025b), which enables continuous evaluation at arbitrary coordinates via cross-attention between query embeddings and encoder features. Query coordinates are first embedded with random Fourier features (Tancik et al., 2020) to form queries $\mathbf{x}_0$, which are updated through $K$ cross-attention blocks with the encoder output $\mathbf{z}_L$:

$$\mathbf{x}_k' = \mathbf{x}_{k-1} + \text{MHA}\big( \text{LN}(\mathbf{x}_{k-1}), \text{LN}(\mathbf{z}_L), \text{LN}(\mathbf{z}_L)\big),$$
$$\mathbf{x}_k = \mathbf{x}_k' + \text{MLP}\big( \text{LN}(\mathbf{x}_k')\big), \quad k = 1,\ldots,K.$$

A lightweight feedforward network then projects the final query embedding $\mathbf{x}_K$ to the target field dimension.

**Latent diffusion with conditional Diffusion Transformer.** We then implement the diffusion process in the latent space using a standard Diffusion Transformer (DiT) (Peebles & Xie, 2023). Conditioning is introduced through noisy sparse solution measurements $\mathbf{y}$, which are encoded by the pretrained FAE encoder $\mathcal{E}_\theta$. The resulting guidance vectors are additively combined with the diffusion inputs $\mathbf{c}$: $\tilde{\mathbf{z}} = \mathbf{z} + \mathcal{E}_\theta(\mathbf{c})$.

**Training.** Our training procedure consists of two stages. In the first stage, we train a GeoFAE to learn a compact and expressive latent representation of functions. For each instance $i$ with geometry $\Omega_i$ and one associated field $u_i$, construct the input features $\mathbf{c}_i$ as defined above and sample query points $\{q_{i,j}\}_{j=1}^M \subset \Omega_i$. The decoder predicts $\hat{u}_i(q_{i,j}) = \mathcal{D}_\phi(\mathcal{E}_\theta(\mathbf{c}_i))(q_{i,j})$. The GeoFAE is trained with the reconstruction loss

$$\mathcal{L}_{\text{FAE}} = \frac{1}{NM} \sum_{i=1}^N \sum_{j=1}^M |u_i(q_{i,j}) - \hat{u}_i(q_{i,j})|^2, \tag{8}$$

where both instances and query points are resampled at each iteration to encourage generalization across geometries and discretizations.

In the second stage, we freeze the pretrained GeoFAE and train a conditional DiT in its latent space. Following the rectified flow framework (Liu et al., 2022), the training objective is

$$\mathcal{L}_{\text{CRF}} = \mathbb{E}_{\substack{(u,\mathbf{c})\sim p_{\text{data}} \\ \mathbf{z}_0 \sim \mathcal{N}(0,\mathbf{I}),\ t\sim\mathcal{U}[0,1]}} \left[ \| (\mathbf{z}_1 - \mathbf{z}_0) - \mathbf{g}_\psi(\mathbf{z}_t, t, \mathbf{z}_c) \|_2^2 \right], \tag{9}$$

where

$$\mathbf{z}_1 = \mathcal{E}_\theta(\mathbf{c}_{\text{ref}}), \quad \mathbf{c}_{\text{ref}} = (V_\Omega, M_{V_\Omega}, u(V_\Omega)), \quad \mathbf{z}_c = \mathcal{E}_\theta(\mathbf{c}), \quad \mathbf{z}_t = (1-t)\mathbf{z}_1 + t\mathbf{z}_0. \tag{10}$$

Here, $\mathbf{z}_1$ is the reference embedding of the full field, $\mathbf{z}_c$ encodes the partial observation. The DiT learns a velocity field $\mathbf{g}_\psi$ that drives samples from noise to the posterior latent distribution conditioned on sparse observations.

**Inference.** At inference, the model generates solution fields conditioned on coarse measurements $\mathbf{C}$. First, the conditioning embedding is obtained via the pretrained GeoFAE encoder $\mathbf{z}_c = \mathcal{E}_\theta(\mathbf{c})$. Next, starting from a Gaussian noise $\mathbf{z}_0 \sim \mathcal{N}(0,\mathbf{I})$, we integrate the learned velocity field to recover the latent code $\mathbf{z}_1$:

$$\frac{d\mathbf{z}(t)}{dt} = \mathbf{g}_\psi(\mathbf{z}(t), t, \mathbf{z}_c), \quad t \in [0,1].$$

Finally, $\mathbf{z}_1$ is passed through the GeoFAE decoder to reconstruct the continuous target field over the geometry, which can be evaluated at arbitrary locations and resolutions.

Table 1: Test errors (relative $L^2$ error, ↓) on five benchmarks. Best results are in **bold**; second best are underlined.

| Model | #Params | Darcy | Cylinder | Plasticity | Airfoil | Ahmed Body |
|---|---|---|---|---|---|---|
| DPS (Interp) | 15M | 0.0514 | 0.1912 | 0.0700 | - | - |
| ViT (Interp) | 15M | 0.0111 | 0.1568 | 0.0292 | - | - |
| UNet (Interp) | 17M | 0.0208 | 0.1561 | 0.0299 | - | - |
| FNO (Interp) | 17M | 0.0091 | 0.1624 | 0.0298 | - | - |
| Geo-FNO | 18M | 0.0065 | 0.1298 | 0.0326 | 0.1094 | 0.2272 |
| Transolver | 17M | 0.0253 | 0.0993 | 0.0172 | 0.0641 | 0.0876 |
| GeoFAE (ours) | 7M | **0.0064** | **0.0538** | **0.0132** | **0.0083** | 0.0820 |
| GeoFunFlow (ours) | 15M | 0.0085 | 0.0567 | 0.0136 | 0.0087 | **0.0811** |

**Theoretical guarantees of posterior approximation.** Here we provide theoretical guarantees showing that GeoFunFlow approximates the true posterior under assumptions. Let $P^*(\cdot \mid \mathbf{c})$ denote the ground truth posterior and $\widehat{P}(\cdot \mid \mathbf{c})$ denote the learned posterior. Under mild assumptions, we establish the following bound:

$$W_2^{\Omega}\Big(\widehat{P}(\cdot \mid \mathbf{c}),\, P^*(\cdot \mid \mathbf{c})\Big) \;\leq\; L_D\, \epsilon_{\text{flow}} \;+\; \epsilon_{\text{rec}}(\mathbf{c}), \tag{11}$$

where $L_D$ is the Lipschitz constant of the GeoFAE decoder, $\epsilon_{\text{flow}}$ is the latent flow error with respect to the ideal latent posterior, and $\epsilon_{\text{rec}}(\mathbf{c})$ is the reconstruction error of the encoder–decoder pair. Intuitively, this result shows that accurate posterior approximation is guaranteed whenever the latent flow is well-trained and the autoencoder provides faithful reconstruction.

Furthermore, when the domain is equipped with $m$ quasi-uniform sensors with fill distance $h_X \sim m^{-1/d}$, and under Sobolev regularity ($s > d/2$) with an $H^s$-Lipschitz decoder, the approximation error further satisfies

$$W_2^{\Omega}\Big(\widehat{P},\, P^*\Big) \;\leq\; C\Big(L_{D,s}\, \epsilon_{\text{flow}} \;+\; h_X^s\big(L_{D,s}\, \epsilon_{\text{flow}} + \epsilon_{\text{rec,s}}(\mathbf{c})\big)\Big), \tag{12}$$

where $C$ is a constant independent of $m$. In particular, increasing the number of sensors ($m \uparrow$) improves posterior accuracy at the rate $h_X^s \sim m^{-s/d}$. This theoretical result matches our empirical results (Figure 4), where the approximation error decreases as more observations are provided. Detailed assumptions, statements, and proofs are given in Appendix B.

## 6 RESULTS

We now demonstrate the effectiveness of our method. We begin by describing the benchmark datasets, selected baselines, and the training and evaluation setup, followed by a presentation of the main results and ablation studies.

**Benchmarks.** We evaluate our method on several PDE benchmarks involving complex geometries, ranging from elliptic PDEs, fluid dynamics, to solid mechanics. The Darcy dataset from Lu et al. (2022) involves porous media flow on triangular domains with a notch, where the objective is to learn the mapping from random boundary conditions to pressure fields. For fluid dynamics, we adopt the Cylinder and Airfoil benchmarks of Pfaff et al. (2020), which simulate incompressible and compressible flows, respectively, on fixed Eulerian meshes. The plasticity dataset introduced by Li et al. (2023b) models elasto-plastic deformation of solids under random die geometries and requires predicting time-dependent displacement fields. Finally, the Ahmed body benchmark tests aerodynamic prediction on complex car geometries using high-resolution CFD simulations. Across all benchmarks, our goal is to train neural networks to reconstruct full solution fields over complex geometries from sparse, noisy measurements. More details are provided in Appendix C.1.

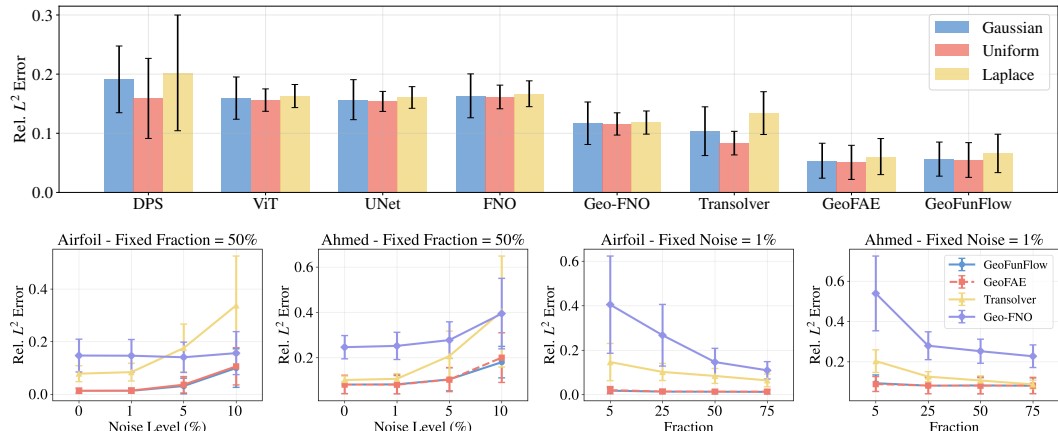

Figure 4: **Top:** Relative $L^2$ error of different baselines under various noise types on the *Cylinder* benchmark, with the fraction fixed at $0.5$ and noise level at $0.01$. **Bottom:** Relative $L^2$ error of different baselines on the *Airfoil* and *Ahmed body* datasets, evaluated by varying the noise level with fixed fraction, or varying the fraction with fixed noise level.

**Baselines.** We evaluate against two complementary classes of baselines. The first consists of models developed for regular domains with uniform grids, including UNet (Ronneberger et al., 2015), ViT (Dosovitskiy et al., 2020), Fourier Neural Operators (FNO) (Li et al., 2021), and diffusion posterior sampling (DPS) (Chung et al., 2022). Since our benchmarks involve complex geometries, we adapt these methods by interpolating data to a square grid for training and mapping predictions back to the original domain for evaluation. The second class comprises models designed specifically for irregular domains, including Geo-FNO (Li et al., 2023b), which incorporates geometric priors into the Fourier neural operator framework, and Transolver (Wu et al., 2024), which employs transformer-based attention over mesh discretizations. A full specification of model architectures and hyperparameters is provided in Appendix C.2.

**Training and evaluation.** We use a unified training recipe for all baselines and benchmarks. We train all models for $10^5$ iterations using the AdamW optimizer (Kingma & Ba, 2014) with a weight decay of $10^{-5}$. The learning rate schedule consists of a linear warm-up over $5,000$ steps from zero to $10^{-3}$, followed by an exponential decay with factor $0.9$ every $5,000$ steps. Batch sizes range from $32$ to $256$, depending on problem size and GPU memory constraints. To improve robustness, inputs are corrupted with Gaussian noise at $1\%$ amplitude and randomly subsampled at fractions $\{0.25, 0.5, 0.75, 1.0\}$. For baselines defined on regular grids, we interpolate the corrupted and masked data to a rectangular domain. The training objective is mean squared error (MSE) between predictions and clean targets.

For problems with varying mesh resolutions, each mesh is subsampled at every epoch to a fixed-size point cloud, enabling mini-batch training. Evaluation is performed using the relative $L^2$ error, averaged across all target variables. For GeoFunFlow, inference is run for $10$ steps. Baselines on regular domains are interpolated back to the original mesh or point cloud to ensure fair comparison.

**Main results.** Table 1 reports reconstruction errors across different baselines and benchmarks, with the sampling fraction fixed at $0.75$ and Gaussian noise at level $1\%$ added to the inputs. Overall, GeoFAE and GeoFunFlow achieve the lowest reconstruction errors among all methods, with GeoFunFlow performing slightly worse than GeoFAE alone. This gap is expected, as the diffusion module is primarily intended to approximate the posterior distribution and provide uncertainty estimates, which may not be able to further reduce reconstruction error.

Figure 1 provides representative reconstructions of the target fields across baselines. GeoFunFlow consistently outperforms other baselines while additionally producing an uncertainty map, quantified as the standard deviation of ensemble predictions from the diffusion model. Notably, the regions of highest uncertainty align with areas where the geometry exhibits potential variations, providing meaningful diagnostic information beyond point predictions. More detailed quantitative results across each variable of interest and additional qualitative comparisons are provided in Figure C.4.

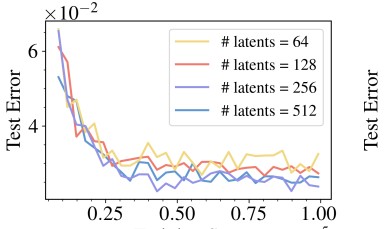 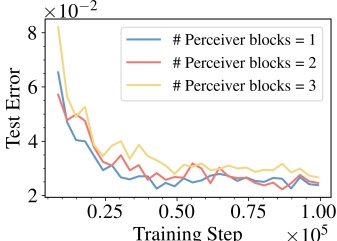 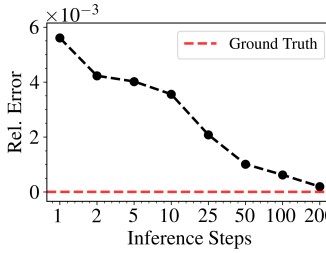

Figure 5: *Ablation studies of GeoFAE during training and inference.* **Left:** Evolution of validation errors with different numbers of latents in the Perceiver module. **Middle:** Evolution of validation errors with different numbers of Perceiver blocks. **Right:** Effect of varying the number of steps for integrating the learned velocity field from the diffusion module. All other hyperparameters are kept at their default values.

An important property of our framework is its robustness to both sparse sampling and noisy measurements. As shown in the top panel of Figure 4, our method consistently achieves the lowest test error across all noise types, demonstrating strong resilience to corrupted inputs. The bottom panel further evaluates robustness on the *Airfoil* and *Ahmed body* benchmarks. Across a wide range of noise levels and sampling fractions, GeoFunFlow reliably outperforms all baselines, maintaining substantially lower reconstruction errors even in out-of-distribution settings, such as noise level of $0.1$ or sampling fraction of $0.05$. These results demonstrate that our approach not only provides accurate posterior estimates under regular conditions, but also remains stable and effective when observations are scarce or severely corrupted—an essential property for real-world scientific and engineering applications.

**Ablation studies.** We perform ablation studies on the *Cylinder* dataset to evaluate the effect of Perceiver hyperparameters in GeoFAE. Specifically, we vary the number of latent variables and the number of Perceiver blocks while keeping all other settings fixed. The resulting test error curves during the training are shown in the left and middle panels of Figure 5. We observe that increasing the number of latents yields only marginal gains, and a single Perceiver block is sufficient since adding more blocks gives comparable accuracy.

Furthermore, we study the sampling efficiency of GeoFunFlow by varying the number of inference steps used to integrate the learned velocity field, with a reference solution obtained from $1000$ steps. As shown in the right panel of Figure 5, the reconstruction error remains low even with very few steps, and even a single-step sampler already provides competitive accuracy. It indicates that GeoFAE learns a well-structured latent representation where the rectified flow trajectory is nearly straight, enabling highly efficient sampling.

# 7 DISCUSSION

We introduced *GeoFunFlow*, a diffusion-based framework for learning inverse operators over complex geometries. The core of our framework is the *GeoFAE*, which integrates a Perceiver module to accommodate varying meshes and point clouds, and employs cross-attention to condition query coordinates for continuous reconstruction. A diffusion model over the latent space of the encoder enables approximation of the full posterior distribution. Across diverse benchmarks, GeoFunFlow achieves state-of-the-art accuracy and demonstrates strong robustness to varying noise levels and sampling sparsity, highlighting its potential for real-world scientific and engineering applications.

Despite these promising results, our work has several limitations that open directions for future research. First, posterior variance remains relatively high in some cases, suggesting that the current design of GeoFAE may not fully exploit geometric structure. Incorporating dedicated geometric encoders could improve representation quality and reduce uncertainty. Second, the framework does not explicitly enforce physical consistency. Integrating physics-informed constraints, either during training or at inference, offers a promising direction to improve both accuracy and reliability. Finally, our experiments are limited to benchmark-scale datasets. Scaling GeoFunFlow to large, realistic problems with millions of mesh elements is an important step toward broader impact.

## ETHICS STATEMENT

This work focuses on methods for solving scientific inverse problems governed by PDEs. The primary applications we target are in physics, engineering, and biomedical domains. We are not aware of any direct ethical concerns beyond those common to scientific machine learning, such as the potential misuse of improved modeling techniques in sensitive domains (e.g., defense applications). We emphasize that our benchmarks are publicly available, non-sensitive, and purely synthetic or simulation-based.

## REPRODUCIBILITY STATEMENT

All datasets used in this work are publicly available and properly cited. Detailed descriptions of the benchmarks, model architectures, training procedures, and hyperparameters are provided in the Appendix. Upon acceptance, we will release our code and pretrained models to ensure full reproducibility.

## USE OF LARGE LANGUAGE MODELS

We used large language models (LLMs) to help refine the writing and presentation of this paper (e.g., clarifying explanations, improving readability, and rephrasing drafts). All technical content—including problem formulation, method design, theoretical results, and experiments—was conceived, implemented, and validated by the authors.

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

# A  NOTATIONS

Table 2 summarizes the main symbols and notations used in this work.

Table 2: Summary of the main symbols and notations used in this work.

| Notation | Description |
|---|---|
| **Operator Learning** | |
| $\Omega \subset \mathbb{R}^d$ | Bounded spatial domain embedded in $d$-dimensional Euclidean space. |
| $\mathcal{U}, \mathcal{V}$ | Input/output function spaces. |
| $\mathcal{G}^\star : \mathcal{U} \to \mathcal{V}$ | Target operator mapping inputs to outputs. |
| $\mathcal{L}_\kappa$ | PDE differential operator parameterized by coefficient field $\kappa$. |
| $\mathcal{B}$ | Boundary operator prescribing conditions on $\partial\Omega$. |
| $\mathcal{S}$ | Sampling/discretization operator producing observations. |
| $(\mathcal{G}_\theta)_{\theta \in \Theta}$ | Parametric neural-operator family indexed by parameters $\theta \in \Theta$. |
| $\widehat{R}_n(\theta)$ | Training objective. |
| **Rectified Flow** | |
| $x_t$ | State along the interpolation/flow at time $t \in [0,1]$. |
| $v_\theta$ | Learned velocity field. |
| $\pi_0$ | The base distribution $\mathcal{N}(0, I)$. |
| $\mathcal{L}_{\mathrm{RF}}(\theta), \mathcal{L}_{\mathrm{CRF}}(\theta)$ | Flow-matching training objective (unconditional/conditional). |
| $p(x \mid y), p_\theta(x \mid y)$ | True/model conditional posterior distribution of $x$ given $y$. |
| **Problem Setup** | |
| $\Lambda = \{\Omega_i\}$ | Family of admissible geometries. |
| $V_\Omega$ | A point cloud discretization of a geometry |
| $\mathcal{F}_\Omega$ | Geometry-dependent forward operator. |
| $X = \{x_i\}_{i=1}^m$ | Pointwise measurement locations. |
| $M_X$ | Mask function on $X$. |
| $\boldsymbol{y}$ | Noisy observations collected at $X$. |
| $\epsilon \sim \mathcal{N}(0, \Sigma)$ | Additive observation noise. |
| $\mathcal{P}_2(\cdot)$ | Probability measures with finite second moment. |
| $W_2^\Omega$ | 2-Wasserstein distance. |
| $\mathcal{C}$ | Conditioning space. |
| $\mathcal{X}_{\mathrm{sens}}$ | Set of admissible sensor locations. |
| $\mathbf{c} = (\Omega, X, \boldsymbol{y})$ | Conditioning instance. |
| **GeoFunFlow** | |
| $\mathbf{z}_q$ | Learnable latent queries. |
| MHA, MSA, LN | Multi-head attention, multi-head self-attention and layer normalization. |
| $\mathcal{E}_\theta, \mathcal{D}_\phi$ | GeoFAE encoder and decoder with parameters $\theta, \phi$. |
| $\mathbf{g}_\psi$ | DiT-parameterized velocity field with parameters $\psi$. |
| **Theoretical Guarantee** | |
| $H^s$ | Sobolev space. |
| $P^*(\cdot \mid \mathbf{c}), \widehat{P}(\cdot \mid \mathbf{c})$ | True and learned posteriors conditioned on $\mathbf{c}$. |
| $L_D$ | Lipschitz constant of the decoder. |
| $h_X$ | Fill distance |
| $\epsilon_{\mathrm{flow}}, \epsilon_{\mathrm{rec}}$ | Latent flow error and reconstruction error. |

## B  THEORETICAL GUARANTEE

We provide a rigorous formulation and proof of the posterior approximation result referenced in the main text. Let $\Omega \subset \mathbb{R}^d$ be a domain and let $\mathcal{U}(\Omega) \subset L^2(\Omega; \mathbb{R}^p)$ be the function space of fields of interest. Recall that a conditioning instance is given by

$$\mathbf{c} = \big(x_i, M(x_i, X), u(x_i) \cdot M(x_i, X)\big)_{i=1}^m,$$

where $X = \{x_i\}_{i=1}^m \subset \Omega$ denotes the set of sensor locations, $M(x_i, X) = \mathbf{1}_{\{x_i \in X\}}$ is the binary mask indicating whether node $x_i$ is observed, and $u(x_i)$ are the corresponding sensor measurements (zero elsewhere). Thus, $\mathbf{c}$ encodes both the geometry of the discretized domain and the sparse, noisy observations available at sensor points.

**Encoder and decoder maps.**    The *encoder* takes only the conditioning instance:

$$\mathcal{E}: \ \mathcal{C} \to Z = \mathbb{R}^\ell, \qquad z_{\mathbf{c}} := \mathcal{E}(\mathbf{c}).$$

The *decoder* evaluates fields pointwise,

$$D: Z \times \Omega \to \mathbb{R}^p, \qquad [T_\Omega(z)](q) := D(z, q),$$

so that $T_\Omega : Z \to L^2(\Omega; \mathbb{R}^p)$ is the decode-to-field operator. Given $\mathbf{c}$, a conditional rectified flow yields a latent distribution $Q_Z(\cdot \mid \mathbf{c}) \in \mathcal{P}_2(Z)$ and the model posterior

$$\widehat{P}(\cdot \mid \mathbf{c}) := (T_\Omega)_\# Q_Z(\cdot \mid \mathbf{c}) \in \mathcal{P}_2\big(L^2(\Omega; \mathbb{R}^p)\big).$$

We also consider the *deterministic reconstruction* $R_{\mathbf{c}} := T_\Omega\big(E(\mathbf{c})\big)$. For any $\Omega$, $W_2^\Omega$ denotes the 2-Wasserstein distance on $\mathcal{P}_2(L^2(\Omega; \mathbb{R}^p))$ induced by $\|\cdot\|_{L^2(\Omega)}$.

**Assumption 1** (Decoder Lipschitzness, reconstruction, and flow accuracy). *For every $\mathbf{c} \in \mathcal{C}$:*

*(i) **Decoder Lipschitzness.** There exists $L_D < \infty$ such that,*

$$\|T_\Omega(z) - T_\Omega(z')\|_{L^2(\Omega)} \le L_D \, \|z - z'\|_2, \qquad \forall z, z' \in Z, \, \Omega \in \Lambda.$$

*(ii) **Reconstruction accuracy.** For $U \sim P^*(\cdot \mid \mathbf{c})$,*

$$\epsilon_{\mathrm{rec}}(\mathbf{c})^2 := \mathbb{E}\Big[\|T_\Omega(E(\mathbf{c})) - U\|_{L^2(\Omega)}^2\Big] < \infty.$$

*(iii) **Latent flow accuracy.** We assume that there exist a $P_Z^*(\cdot \mid \mathbf{c}) \in \mathcal{P}_2(Z)$ such that*

$$(T_\Omega)_\# \, P_Z^*(\cdot \mid \mathbf{c}) = P^*(\cdot \mid \mathbf{c}),$$

*and*

$$W_2(Q_Z(\cdot \mid \mathbf{c}), \, P_Z^*(\cdot \mid \mathbf{c})) \le \epsilon_{\mathrm{flow}}.$$

**Remark.** *Since $T_\Omega : Z \to L^2(\Omega; \mathbb{R}^p)$ is Borel measurable between Polish spaces, standard measurable selection results (e.g., Kuratowski-Ryll-Nardzewski) guarantee the existence of a latent posterior $P_Z^*(\cdot \mid \mathbf{c}) \in \mathcal{P}_2(Z)$ such that*

$$(T_\Omega)_\# \, P_Z^*(\cdot \mid \mathbf{c}) = P^*(\cdot \mid \mathbf{c}).$$

**Lemma 1** (Stability of $W_2$ under Lipschitz maps). *Let $T : Z \to \mathsf{X}$ be $L$-Lipschitz into a Hilbert space $(\mathsf{X}, \|\cdot\|)$, and let $P, Q \in \mathcal{P}_2(Z)$. Then*

$$W_2\big(T_\# P, \ T_\# Q\big) \le L \, W_2(P, Q).$$

*Proof.* Let $\pi$ be an optimal coupling of $P$ and $Q$. Then $(T, T)_\# \pi$ couples $T_\# P$ and $T_\# Q$, and

$$\int \|T(z) - T(z')\|^2 \, d\pi(z, z') \le L^2 \int \|z - z'\|^2 \, d\pi(z, z') = L^2 W_2(P, Q)^2.$$

Taking the infimum over couplings on the left yields the result. $\qquad \square$

**Lemma 2** (Moment finiteness of the decoded latent pushforward). *Under Assumption 1(ii), if $U \sim P^*(\cdot \mid \mathbf{c})$ and $\widetilde{U} := T_\Omega(E(\mathbf{c}))$, then $\mathbb{E}\|\widetilde{U}\|^2_{L^2(\Omega)} < \infty$, hence $(T_\Omega)_\# P_Z^* \in \mathcal{P}_2(L^2(\Omega; \mathbb{R}^p))$.*

*Proof.* By $\|a + b\|^2 \leq 2\|a\|^2 + 2\|b\|^2$,

$$\mathbb{E}\|\widetilde{U}\|^2_{L^2} \leq 2\,\mathbb{E}\|U\|^2_{L^2} + 2\,\mathbb{E}\|\widetilde{U} - U\|^2_{L^2} = 2\,\mathbb{E}\|U\|^2_{L^2} + 2\,\epsilon_{\mathrm{rec}}(\mathbf{c})^2 < \infty,$$

since $P^*(\cdot \mid \mathbf{c}) \in \mathcal{P}_2(L^2)$ and Assumption 1(ii) holds. $\qquad\square$

**Theorem 3** (Posterior approximation via conditional rectified flow). *Fix $\Omega$ and $\mathbf{c} \in \mathcal{Y}_\Omega$. Under Assumption 1,*

$$W_2^\Omega\Big(\widehat{P}(\cdot \mid \mathbf{c}),\, P^*(\cdot \mid \mathbf{c})\Big) \;\leq\; L_D\, \epsilon_{\mathit{flow}} \;+\; \epsilon_{\mathit{rec}}(\mathbf{c}). \tag{13}$$

*Proof.* By Lemma 2, $(T_\Omega)_\# P_Z^* \in \mathcal{P}_2(L^2(\Omega))$. Then

$$W_2^\Omega\Big(\widehat{P},\, P^*\Big) \;\leq\; W_2^\Omega\big((T_\Omega)_\# Q_Z,\, (T_\Omega)_\# P_Z^*\big) \;+\; W_2^\Omega\big((T_\Omega)_\# P_Z^*,\, P^*\big).$$

For the first term, Lemma 1 with $T = T_\Omega$ gives

$$W_2^\Omega\big((T_\Omega)_\# Q_Z, (T_\Omega)_\# P_Z^*\big) \leq L_D(\Omega)\, W_2(Q_Z, P_Z^*) \leq L_D(\Omega)\, \epsilon_{\mathrm{flow}}. \tag{14}$$

For the second, couple $U \sim P^*$ with $\widetilde{U} := T_\Omega(E(\mathbf{c})) \sim (T_\Omega)_\# P_Z^*$; then

$$W_2^\Omega\big((T_\Omega)_\# P_Z^*,\, P^*\big)^2 \;\leq\; \mathbb{E}\|\,\widetilde{U} - U\,\|^2_{L^2(\Omega)} \;=\; \epsilon_{\mathrm{rec}}(\mathbf{c})^2.$$

Combining the two bounds gives the desired results. $\qquad\square$

Let $X = \{x_i\}_{i=1}^m \subset \Omega$ denote $m$ sensors, and define the (normalized) discrete $\ell^2$ sensor norm

$$\|u\|^2_{X,2} \;:=\; \frac{|\Omega|}{m} \sum_{i=1}^m |u(x_i)|^2, \qquad u: \Omega \to \mathbb{R}^p.$$

Let the *fill distance* be $h_X := \sup_{x \in \Omega} \min_{x_i \in X} \|x - x_i\|_2$ and the *separation radius* $q_X := \frac{1}{2}\min_{i \neq j}\|x_i - x_j\|_2$. We say $X$ is *quasi-uniform* if $h_X \asymp q_X$ with constants independent of $m$ (e.g., uniform grids or blue-noise sets). Note that $h_X \sim m^{-1/d}$ for quasi-uniform $X$.

**Assumption 2** (Sobolev regularity and decoder $H^s$-Lipschitzness). *Fix $s > d/2$. For every $\mathbf{c} \in \mathcal{Y}_\Omega$:*

*(i) **Decoder $H^s$-Lipschitzness.** There exists $L_{D,s} < \infty$ such that*

$$\|T_\Omega(z) - T_\Omega(z')\|_{H^s(\Omega)} \;\leq\; L_{D,s}\, \|z - z'\|_2, \qquad \forall z, z' \in Z.$$

*(ii) **Finite $H^s$-reconstruction error.** Define the $H^s$ reconstruction error*

$$\epsilon_{\mathrm{rec,s}}(\mathbf{c})^2 := \mathbb{E}\Big[\|T_\Omega(E(\mathbf{c})) - U\|^2_{H^s(\Omega)}\Big] < \infty.$$

**Lemma 4** (Posterior smoothness from decoder regularity and finite moments). *Fix $s > \frac{d}{2}$ and $\mathbf{c} \in \mathcal{Y}_\Omega$.*

*Then both posteriors are supported on $H^s(\Omega; \mathbb{R}^p)$ and have finite $H^s$-second moments, i.e.*

$$\mathbb{E}\big[\|U\|^2_{H^s} \,\big|\, \mathbf{c}\big] < \infty \quad (U \sim P^*), \qquad \mathbb{E}\big[\|\widehat{U}\|^2_{H^s} \,\big|\, \mathbf{c}\big] < \infty \quad (\widehat{U} \sim \widehat{P}).$$

*Proof.* **Model posterior.** Let $Z \sim Q_Z(\cdot \mid \mathbf{c})$ and $\widehat{U} := T_\Omega(Z)$. By $H^s$-Lipschitzness and $T_\Omega(0) \in H^s$,

$$\|\widehat{U}\|_{H^s} \leq \|T_\Omega(Z) - T_\Omega(0)\|_{H^s} + \|T_\Omega(0)\|_{H^s} \leq L_{D,s}\|Z\|_2 + \|T_\Omega(0)\|_{H^s}.$$

Hence $\mathbb{E}\|\widehat{U}\|^2_{H^s} \leq 2L_{D,s}^2\,\mathbb{E}\|Z\|^2_2 + 2\|T_\Omega(0)\|^2_{H^s} < \infty$ since $Q_Z \in \mathcal{P}_2(Z)$.

**True posterior.** Let $U \sim P^*(\cdot \mid \mathbf{c})$, set $Z^* := E(\mathbf{c})$ and $\widetilde{U} := T_\Omega(Z^*)$. Then

$$\|U\|_{H^s} \leq \|U - \widetilde{U}\|_{H^s} + \|\widetilde{U}\|_{H^s}.$$

Squaring and using $(a+b)^2 \le 2a^2 + 2b^2$,

$$\mathbb{E}\|U\|_{H^s}^2 \le 2\,\mathbb{E}\|U - \widetilde{U}\|_{H^s}^2 + 2\,\mathbb{E}\|\widetilde{U}\|_{H^s}^2 = 2\,\epsilon_{\mathrm{rec,s}}(\mathbf{c})^2 + 2\,\mathbb{E}\|\widetilde{U}\|_{H^s}^2.$$

Since $Z^* \sim P_Z^*(\cdot \mid \mathbf{c}) \in \mathcal{P}_2(Z)$, the same $H^s$-Lipschitz bound as above yields $\mathbb{E}\|\widetilde{U}\|_{H^s}^2 < \infty$. Thus $\mathbb{E}\|U\|_{H^s}^2 < \infty$. $\qquad\square$

**Lemma 5** (Sampling inequality for $H^s$, $s > d/2$). *Let $s > d/2$ and $X \subset \Omega$ be quasi-uniform with fill distance $h_X$. There exists $C = C(\Omega, s)$ such that, for all $u \in H^s(\Omega; \mathbb{R}^p)$,*

$$\|u\|_{L^2(\Omega)} \;\le\; C\Big( \|u\|_{X,2} \;+\; h_X^s\,|u|_{H^s(\Omega)} \Big).$$

**Remark.** *Lemma 5 is standard in scattered data approximation / Marcinkiewicz–Zygmund–type sampling inequalities. For quasi-uniform $X$, $h_X \sim m^{-1/d}$, so the second term decays like $m^{-s/d}$.*

**Theorem 6** (Posterior error with $m$ sensors). *Under Assumptions 1 and 2 with $s > d/2$, and for quasi-uniform $X$ with fill distance $h_X$, there exists $C = C(\Omega, s)$ such that*

$$W_2^\Omega(\widehat{P}, P^*) \;\le\; C\,L_{D,s}\,\epsilon_{\mathrm{flow}} \;+\; C\,h_X^s\Big( L_{D,s}\,\epsilon_{\mathrm{flow}} + \epsilon_{\mathrm{rec,s}}(\mathbf{c}) \Big).$$

*Proof.* Let $\pi$ be any coupling of $\widehat{P}$ and $P^*$ on $L^2(\Omega)$, and write pairs as $(\widehat{U}, U)$. By Lemma 5,

$$\|\widehat{U} - U\|_{L^2} \;\le\; C\Big( \|\widehat{U} - U\|_{X,2} + h_X^s\,|\widehat{U} - U|_{H^s} \Big).$$

Taking expectations w.r.t. $\pi$ and minimizing over $\pi$ yields

$$W_2^\Omega(\widehat{P}, P^*) \;\le\; C\Big( \underbrace{\inf_\pi \big(\mathbb{E}_\pi\|\widehat{U} - U\|_{X,2}^2\big)^{1/2}}_{\mathcal{T}_1} + h_X^s \underbrace{\inf_\pi \big(\mathbb{E}_\pi|\widehat{U} - U|_{H^s}^2\big)^{1/2}}_{\mathcal{T}_2} \Big).$$

*Bounding $\mathcal{T}_1$.* Couple via latent variables: draw $(Z, Z^*)$ optimally for $(Q_Z, P_Z^*)$ and set $\widehat{U} = T_\Omega(Z)$, $\widetilde{U} = T_\Omega(Z^*)$. Then

$$\|\widehat{U} - \widetilde{U}\|_{X,2} \;\le\; \|\widehat{U} - \widetilde{U}\|_{L^\infty(\Omega)} \;\lesssim\; \|\widehat{U} - \widetilde{U}\|_{H^s(\Omega)} \;\le\; L_{D,s}\,\|Z - Z^*\|_2,$$

using $s > d/2$ (Sobolev embedding $H^s \hookrightarrow L^\infty$) and decoder $H^s$-Lipschitzness. Hence

$$\mathcal{T}_1 \;\lesssim\; L_{D,s}\,W_2(Q_Z, P_Z^*) \le L_{D,s}\,\epsilon_{\mathrm{flow}}. \tag{15}$$

*Bounding $\mathcal{T}_2$.* Insert $\widetilde{U} := T_\Omega(E(\mathbf{c}))$ and split $|\widehat{U} - U|_{H^s} \le |\widehat{U} - \widetilde{U}|_{H^s} + |\widetilde{U} - U|_{H^s}$. With the same latent coupling as above,

$$\big(\mathbb{E}|\widehat{U} - \widetilde{U}|_{H^s}^2\big)^{1/2} \le L_{D,s}\,W_2(Q_Z, P_Z^*) \le L_{D,s}\,\epsilon_{\mathrm{flow}},$$

and, by definition, $\big(\mathbb{E}|\widetilde{U} - U|_{H^s}^2\big)^{1/2} = \epsilon_{\mathrm{rec,s}}(\mathbf{c})$. Thus $\mathcal{T}_2 \le L_{D,s}\,\epsilon_{\mathrm{flow}} + \epsilon_{\mathrm{rec,s}}(\mathbf{c})$, which completes the bound. $\qquad\square$

# C EXPERIMENTS

## C.1 BENCHMARKS

We provide details of the benchmarks used in our experiments, with a summary in Table 1. We note that some existing benchmarks for neural operators on complex geometries—such as Elasticity (Li et al., 2023b), Pipe, ShapeNet-Car (Chang et al., 2015), AirfRANS (Bonnet et al., 2022), and DrivAerNet (Elrefaie et al., 2025) —are not applicable in our setting. In these cases, the solution is uniquely determined by the geometry itself, meaning that additional observations provide no useful information beyond the geometry. Consequently, they cannot serve as inverse problem benchmarks that should require models to reconstruct fields from sparse observations.

**Darcy.** We follow the setup of Lu et al. (2021) with $K(x, y) = 0.1$ and $f = -1$. Gaussian process samples are used to generate random boundary conditions on triangular domains with a notch. The dataset consists of 2,000 samples on a fixed mesh of 2,295 nodes; each sample has distinct boundary conditions but identical geometry. We use 1,800 samples for training and the remaining 200 for testing.

**Cylinder.** We adopt the Cylinder dataset from Pfaff et al. (2020), which simulates incompressible fluid flow around a cylinder on a 2D Eulerian mesh. Each node contains momentum samples $\mathbf{w}_i$, velocity $\mathbf{u}_i$, and boundary indicators $\mathbf{n}_i$. The flow varies with cylinder size and position, with discretized on meshes of varying sizes of $\mathcal{O}(10^3)$. The dataset consists of 1,000 training samples and 100 test samples, each comprising 600 time steps with step size $\Delta t = 0.01$. The learning task is to reconstruct the full velocity and pressure fields at each snapshot from sparse, noisy measurements.

**Plasticity.** We use the Plasticity dataset generated by Li et al. (2023b), which simulates elasto-plastic forging of a material block $\Omega = [0, L] \times [0, H]$ impacted by a rigid die moving at constant speed $v$. The die shape $S_d$ is randomized via spline interpolation of sampled points. The governing equations follow an elasto-plastic constitutive law with yield stress $\sigma_Y = 70$ MPa, Young's modulus $E = 200$ GPa, Poisson's ratio $\nu = 0.3$, and density $\rho^s = 7850, \mathrm{kg/m^3}$. The dataset contains 900 training samples and 100 test samples, generated using ABAQUS with 3,000 quadrilateral elements. Each solution operator maps the die geometry to time-dependent deformation fields on a $101 \times 31$ mesh over 20 time steps. The learning task is to reconstruct the full field at each snapshot from sparse, noisy measurements.

**Airfoil.** We adopt the Airfoil dataset from Pfaff et al. (2020), which simulates compressible aerodynamics around airfoil cross-sections. The 2D Eulerian mesh encodes momentum $\mathbf{w}$, density $\rho$, and pressure $p$. The flow varies across different airfoil geometries, with each sample discretized on meshes of varying sizes of $\mathcal{O}(10^3)$. The dataset consists of 1,000 training samples and 100 test samples, each comprising 600 time steps with step size $\Delta t = 0.01$. The learning task is to reconstruct the full velocity and pressure fields at each snapshot from sparse, noisy measurements. For this benchmark, we cannot test grid-based baselines such as UNet and FNO due to large interpolation errors.

**Ahmed body.** We adopt the dataset generated by Li et al. (2024c), which simulates turbulent flow over more than 500 Ahmed body car geometries at varying Reynolds numbers using GPU-based OpenFOAM. Each sample is discretized on surface meshes with $\mathcal{O}(10^5)$ nodes and volumetric CFD grids with up to $\mathcal{O}(10^7)$ points. The learning task is to reconstruct the full pressure field from sparse, noisy measurements. We use 500 samples for training and the remaining 51 samples for testing. For this benchmark, we cannot test grid-based baselines such as UNet and FNO due to large interpolation error.

Table 3: Summary of PDE benchmarks used in our experiments, including solver, mesh type, mesh size, temporal resolution (# steps), train/test splits, and field variables of interest.

| Dataset | Solver | Mesh type | Meshing | Mesh size | # steps | Splits (train/test) | Fields |
|---------|--------|-----------|---------|-----------|---------|---------------------|--------|
| Darcy | Matlab PDE | Triangle 2D | irregular | ~2,000 | Steady | 1800 / 200 | $p$ |
| Cylinder | COMSOL | Triangle 2D | irregular | ~2,000 | 600 | 1000 / 100 (traj.) | $u, v, p$ |
| Plasticity | ABAQUS | Quadrilateral 2D | dynamic | ~3,000 | 20 | 900 / 80 (traj.) | $u, v$ |
| Airfoil | SU2 | Triangle 2D | irregular | ~5,000 | 600 | 1000 / 100 (traj.) | $u, v, p$ |
| Ahmed Body | OpenFOAM | Triangle 3D | irregular | ~50,000 | Steady | 500 / 51 | $p$ |

## C.2 MODEL DETAILS

We provide a brief summary of each baseline and the hyperparameters used in our experiments. Hyperparameters are tuned so that all baselines have comparable model sizes for a fair comparison.

### C.2.1 GEOFUNFLOW

**GeoFAE.** For the GeoFAE, the Perceiver uses an embedding dimension of 256 and encodes coordinates with Fourier frequencies initialized by Gaussians $\mathcal{N}(0, 10)$. The number of the trainable latent queries is fixed as 256. The encoder consists of 8 transformer blocks with 8 attention heads and a MLP ratio of 2. The decoder consists of 4 transformer blocks with 8 attention heads and an MLP ratio of 2, where the coordinate embedding is the same as the Perceiver. Finally, a one-layer MLP is employed to project the final query embedding to the target field dimension.

**GeoFunFlow.** For the diffusion part of GeoFunFlow, the embedding dimension of DiTs is 256, matching the latent dimension of GeoFAE. The DiTs consist of 8 transformer blocks with 8 attention heads and a MLP ratio of 2.

### C.2.2 INTERPOLATION-BASED BASELINES

For comparison on irregular point cloud data, we implement grid-based baseline models (ViT, UNet, FNO, DPS) using an interpolation pipeline. We first interpolate each input point cloud to a regular $128 \times 128$ grid using an exponential distance weighting with parameter $\beta$, which assigns each grid cell a weighted average of nearby point values while limiting oversmoothing. We then apply the baseline model on the gridded field and finally interpolate the output back to the query coordinates.

Interpolation error is empirically negligible on benchmarks with smooth fields and near-uniform sampling (Darcy, Cylinder, Plasticity). In contrast, Airfoil exhibits highly uneven sampling and Ahmed-body points lie on 3D surfaces, where interpolation error is non-negligible; therefore we omit interpolation-based baselines for these two benchmarks.

**Vision transformer (ViT).** For ViT, it patchifies the input tensor first with a patch size $16 \times 16$, and embeds each patch token with the embedding dimension of 384. Then it uses 12 transformer blocks with 8 attention heads and a MLP ratio of 2. Finally, a patchwise decoder maps the final tokens back to $128 \times 128$ to reconstruct the output field at the original resolution.

**UNet.** We adopt a standard 2-D UNet which consists of 4 levels of downsampling and expansion processes with a downsampling ratio 2, where the maximum width is 48. In each level, a 2-layer CNN is employed to downsample or expand the shape, and a one-layer CNN is used to incorporate skip connections

**Fourier neural operator (FNO).** The Fourier Neural Operator (FNO) applies a Fast Fourier Transform (FFT) along spatial dimensions, multiplies learned complex-valued weights on a limited number of Fourier modes to perform global convolutions in the frequency domain. We implement a 2-D FNO with width 32, using 32 Fourier modes per dimension across 4 spectral layers.

**Diffusion posterior sampling (DPS).** Diffusion Posterior Sampling (DPS) formulates reconstruction as sampling from $p(x \mid y)$, where $y$ are observations and $x$ is the true field. DPS uses a pre-trained DDPM as a prior for $p(x)$, and modifies the reverse diffusion by a guidance term depending on $y$. We adopt a conditional 4-level UNet with the width of 64 as the backbone of DDPM, pre-trained on the interpolated dataset. During inference, a time-dependent scaling of $\nabla_{x_t} \log p(y \mid x_t)$ is added at each of the 1000 denoising steps, steering the trajectory toward consistency with the measurements.

While DPS is designed for inverse problems with partial observations, its standard formulation assumes observations on a regular grid. In our setting with irregular point-cloud observations, DPS cannot be applied directly; instead, the observations must first be interpolated to a grid. We therefore treat DPS as an interpolation-based baseline and instantiate it within the same interpolation pipeline described above.

### C.2.3 Geometry-based baselines

We also evaluate models designed for complex geometries with point-cloud inputs, which operate natively without interpolation. We include two competitive, representative methods: Geo-FNO and Transolver.

**Geo-FNO.** Geo-FNO extends the Fourier Neural Operator to arbitrary geometries by learning a coordinate transformation that maps the irregular input domain into a latent regular grid, then an FNO-2D or FNO-3D can be applied. Geo-FNO uses a positional encoding of point coordinates to project input into a latent grid of $40 \times 40$ (2-D) or $24 \times 24 \times 12$ (3-D). Then we use an FNO-2D with 20 modes or and FNO-3D with 10 modes to process the data in the projected domain. The number of FNO layers is 5, following Li et al. (2023b). Finally, an inverse deformation is applied to project the output back onto the querying geometry.

**Transolver.** Transolver is a transformer-based neural PDE solver for unstructured meshes and point clouds. It learns Physics-Attention layers to group the multitude of mesh points into a smaller set of latent tokens, representing distinct physical states or regions. In our implementation, each input coordinate with its values and mask embedded into a $512$-dimensional feature. The model has 12 Transolver block, each consisting of a learnable grouping aggregation and a self-attention with 8 heads. The grouping aggregation will aggregate points to 8 latent slices, where each slice can be seen as a cluster of points that likely share a similar physical behavior. Then the self-attention will learn global interactions across tokens. We follow the configuration of with minor tuning to match our datasets.

### C.3 Computational Costs

Table 4 reports the training time for all baselines in each benchmark. All experiments are conducted on a single NVIDIA H200 GPU.

Table 4: Training time (hours; lower is better) across datasets. A dash indicates not applicable.

| Model | Darcy | Cylinder | Plasticity | Airfoil | Ahmed Body |
|---|---|---|---|---|---|
| DPS | 16.5 | 24.0 | 14.6 | — | — |
| ViT | 2.1 | 4.5 | 1.5 | — | — |
| UNet | 11.0 | 22.9 | 10.4 | — | — |
| FNO | 2.8 | 3.3 | 1.4 | — | — |
| Geo-FNO | 15.6 | 6.7 | 15.2 | 5.8 | 17.5 |
| Transolver | 13.8 | 7.4 | 28.5 | 14.5 | 53.5 |
| GeoFAE | 17.4 | 18.9 | 16.2 | 16.3 | 32.7 |
| GeoFunFlow | 15.4 | 15.0 | 15.3 | 15.7 | 29.9 |

### C.4 Additional Results and Visualizations

**Darcy.** Table 5 reports the average test errors and standard deviations across baselines, while Figure 6 illustrates their error distributions over individual test samples. GeoFAE achieves the lowest mean error, slightly outperforming Geo-FNO, while GeoFunFlow remains competitive despite operating in a generative setting. Figure 7 shows a representative reconstruction from sparse and noisy observations. In this example, Geo-FNO fails to recover the correct pressure distribution and Transolver produces non-smooth results, whereas both GeoFAE and GeoFunFlow accurately reconstruct the underlying field.

**Plasticity.** Table 6 reports the average test errors for displacement components $u$ and $v$, while Figure 8 shows the corresponding error distributions across test samples. GeoFAE achieves the

Table 5: Test errors on the Darcy dataset with standard deviations. The best results are shown in **bold**, and the second-best are underlined.

| Model | #Params | Pressure |
|---|---|---|
| DPS (Interp) | 15M | $0.0514 \pm 0.0195$ |
| ViT (Interp) | 15M | $0.0111 \pm 0.0066$ |
| UNet (Interp) | 17M | $0.0208 \pm 0.0108$ |
| FNO (Interp) | 17M | $0.0091 \pm 0.0053$ |
| Geo-FNO | 18M | $\underline{0.0065 \pm 0.0031}$ |
| Transolver | 17M | $0.0253 \pm 0.0107$ |
| GeoFAE (ours) | 7M | $\mathbf{0.0064 \pm 0.0082}$ |
| GeoFunFlow (ours) | 15M | $0.0105 \pm 0.0104$ |

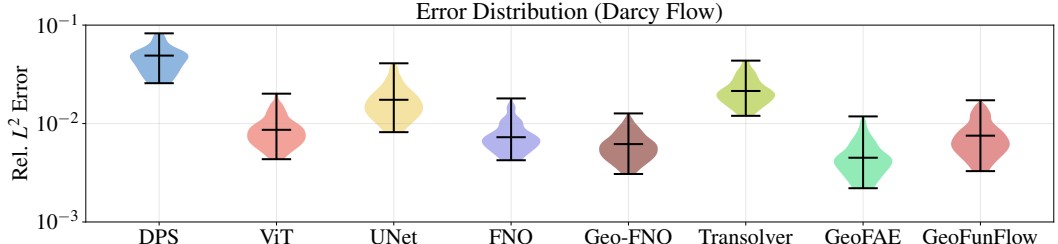

Figure 6: Violin plots of test errors on the Darcy dataset.

lowest errors on both components, with GeoFunFlow performing comparably despite operating in a generative setting. Both methods significantly outperform interpolation-based baselines and established operator-learning models such as FNO and Geo-FNO. Representative reconstructions for $u$ and $v$ are shown in Figures 9 and 10. In these examples, GeoFAE and GeoFunFlow accurately capture fine-scale displacement patterns, whereas several baselines either fail to reconstruct the field or produce nonsmooth, oscillatory predictions.

Table 6: Test errors on the Plasticity dataset with standard deviations. The best results are shown in **bold**, and the second-best are underlined.

| Model | #Params | $U$ | $V$ |
|---|---|---|---|
| DPS (Interp) | 15M | $0.0670 \pm 0.0318$ | $0.0730 \pm 0.0258$ |
| ViT (Interp) | 15M | $0.0212 \pm 0.0118$ | $0.0371 \pm 0.0157$ |
| UNet (Interp) | 17M | $0.0205 \pm 0.0121$ | $0.0392 \pm 0.0164$ |
| FNO (Interp) | 17M | $0.0217 \pm 0.0121$ | $0.0379 \pm 0.0160$ |
| Geo-FNO | 18M | $0.0336 \pm 0.0137$ | $0.0315 \pm 0.0109$ |
| Transolver | 17M | $0.0190 \pm 0.0092$ | $0.0153 \pm 0.0057$ |
| GeoFAE (ours) | 7M | $\mathbf{0.0153 \pm 0.0073}$ | $\mathbf{0.0110 \pm 0.0025}$ |
| GeoFunFlow (ours) | 15M | $\underline{0.0161 \pm 0.0077}$ | $\underline{0.0111 \pm 0.0025}$ |

**Cylinder.** Table 7 reports the average test errors across velocity components $u$, $v$, and pressure, while Figure 11 shows their error distributions over individual samples. GeoFAE achieves the lowest

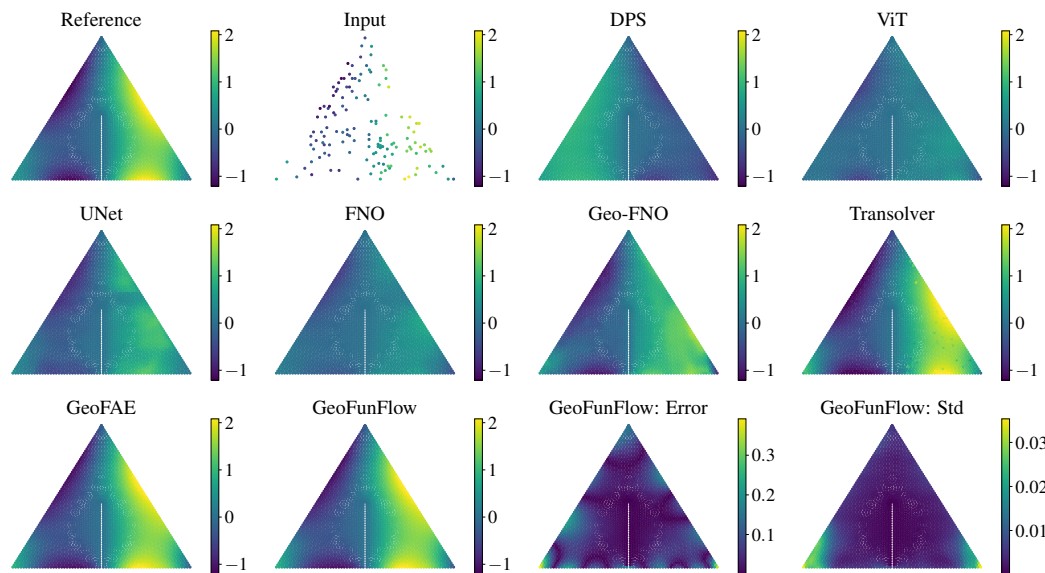

Figure 7: Representative reconstruction on the Darcy dataset.

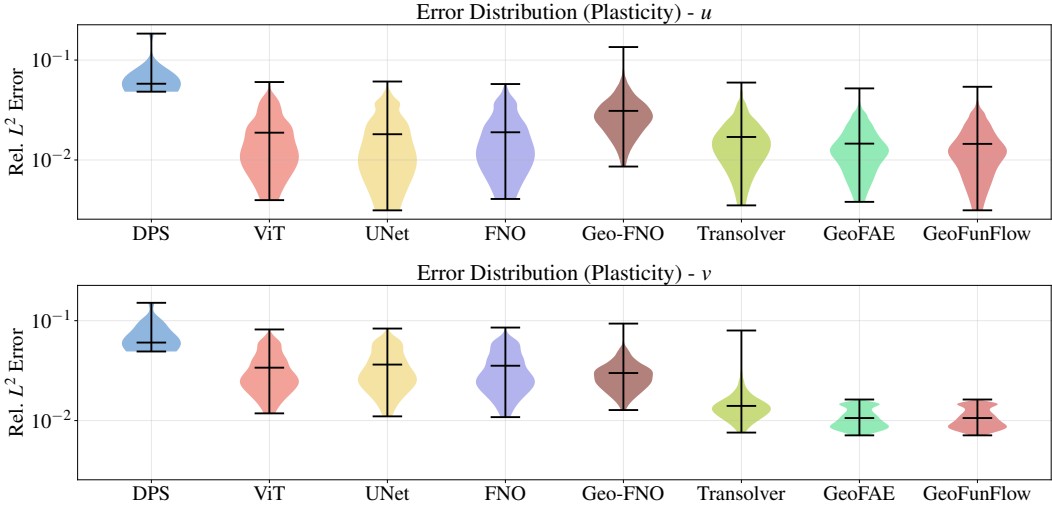

Figure 8: Violin plots of test errors for the Plasticity dataset across displacement components $u$ and $v$.

mean errors across all variables, with GeoFunFlow performing comparably despite its generative formulation.

**Airfoil.** Table 8 reports the average test errors across velocity components $u$, $v$, and pressure, while Figure 12 shows their error distributions over individual samples. GeoFAE achieves the lowest mean errors across all variables, with GeoFunFlow performing comparably despite its generative formulation.

**Ahmed Body.** Table 7 reports the average test errors across velocity components $u$, $v$, and pressure, while Figure 11 shows their error distributions over individual samples. GeoFAE achieves the lowest mean errors across all variables, with GeoFunFlow performing comparably despite its generative formulation. Both methods significantly outperform interpolation-based baselines and operator-learning approaches such as Geo-FNO and Transolver.

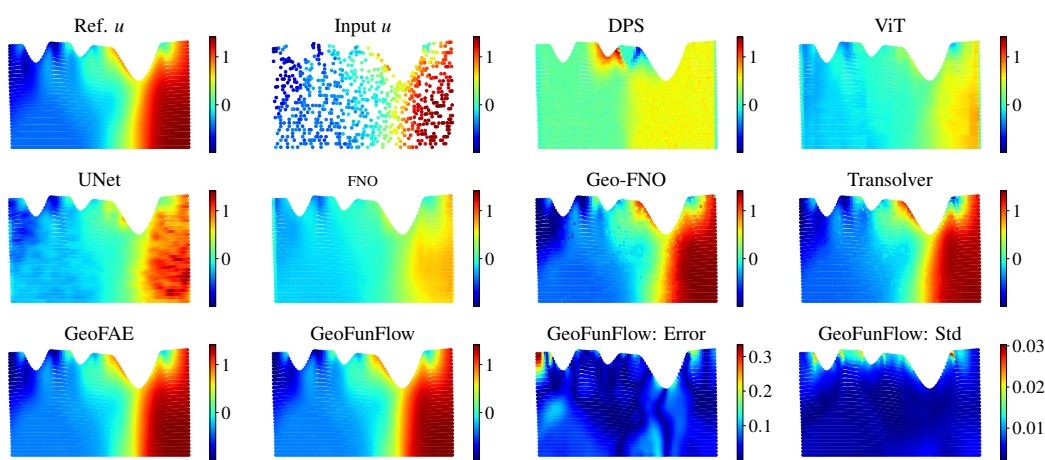

Figure 9: Representative reconstruction of the displacement field $u$ on the Plasticity dataset.

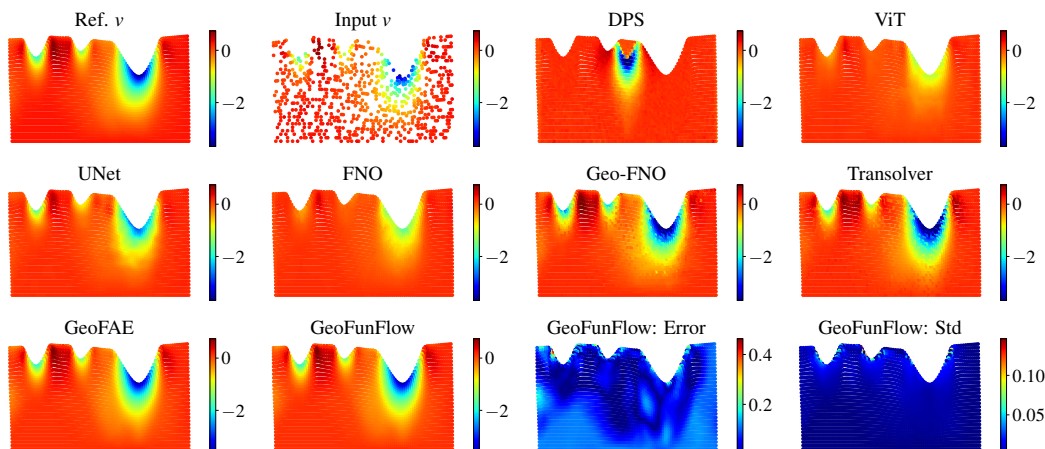

Figure 10: Representative reconstruction of the displacement field $v$ on the Plasticity dataset.

Table 7: Test errors on the Cylinder dataset with standard deviations. The best results are shown in **bold**, and the second-best are underlined.

| Model | #Params | $U$ | $V$ | Pressure |
|-------|---------|-----|-----|----------|
| DPS (Interp) | 15M | $0.1140 \pm 0.0191$ | $0.2997 \pm 0.0545$ | $0.1598 \pm 0.1648$ |
| ViT (Interp) | 15M | $0.1096 \pm 0.0150$ | $0.2719 \pm 0.0322$ | $0.0889 \pm 0.0436$ |
| UNet (Interp) | 17M | $0.1104 \pm 0.0151$ | $0.2660 \pm 0.0285$ | $0.0920 \pm 0.0368$ |
| FNO (Interp) | 17M | $0.1218 \pm 0.0186$ | $0.2675 \pm 0.0350$ | $0.0979 \pm 0.0483$ |
| Geo-FNO | 18M | $0.0767 \pm 0.0177$ | $0.1408 \pm 0.0477$ | $0.1718 \pm 0.0541$ |
| Transolver | 17M | $0.0428 \pm 0.0155$ | $0.1398 \pm 0.0516$ | $0.1154 \pm 0.0624$ |
| GeoFAE (ours) | 7M | $\mathbf{0.0275 \pm 0.0300}$ | $\mathbf{0.0888 \pm 0.0971}$ | $\mathbf{0.0451 \pm 0.0503}$ |
| GeoFunFlow (ours) | 15M | $\underline{0.0299 \pm 0.0299}$ | $\underline{0.0933 \pm 0.0970}$ | $\underline{0.0470 \pm 0.0480}$ |

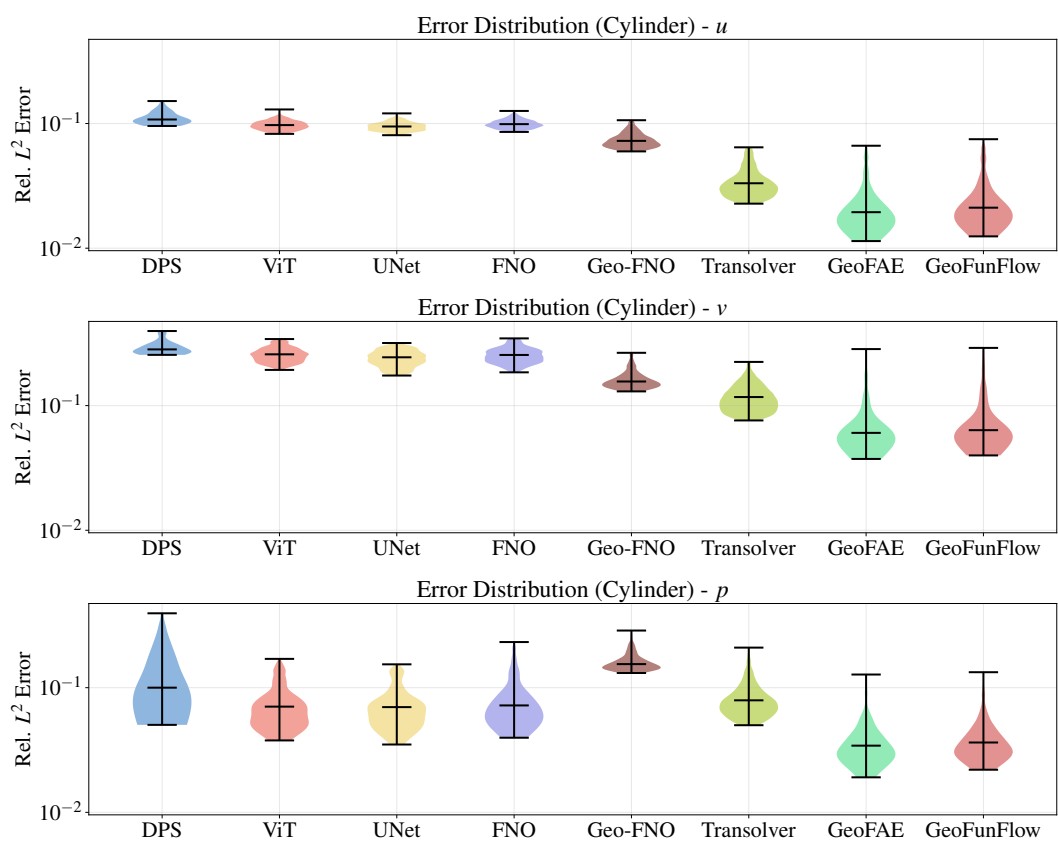

Figure 11: Violin plots of test errors for the Cylinder dataset across the velocity fields $u, v$ and pressure.

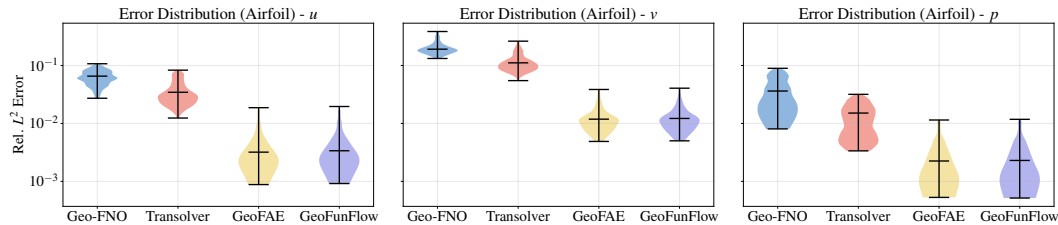

Figure 12: Violin plots of test errors for the Aifoil dataset across the velocity fields $u, v$ and pressure.

Table 8: Test errors on the Airfoil dataset with standard deviations. The best results are shown in **bold**, and the second-best are underlined.

| Model | #Params | $U$ | $V$ | Pressure |
|---|---|---|---|---|
| Geo-FNO | 18M | $0.0698 \pm 0.0216$ | $0.2154 \pm 0.0688$ | $0.0429 \pm 0.0260$ |
| Transolver | 17M | $0.0430 \pm 0.0213$ | $0.1327 \pm 0.0558$ | $0.0165 \pm 0.0088$ |
| GeoFAE (ours) | 7M | $\underline{0.0057 \pm 0.0060}$ | $\underline{0.0156 \pm 0.0111}$ | $\underline{0.0037 \pm 0.0040}$ |
| GeoFunFlow (ours) | 15M | $\mathbf{0.0060 \pm 0.0062}$ | $\mathbf{0.0162 \pm 0.0115}$ | $\mathbf{0.0038 \pm 0.0041}$ |

Table 9: Test errors on the Ahmbed body dataset with standard deviations. The best results are shown in **bold**, and the second-best are underlined.

| Model | #Params | Pressure |
|---|---|---|
| Geo-FNO | 18M | $0.2272 \pm 0.0562$ |
| Transolver | 17M | $0.0876 \pm 0.0213$ |
| GeoFAE (ours) | 7M | $0.0820 \pm 0.0409$ |
| GeoFunFlow (ours) | 15M | $\mathbf{0.0811 \pm 0.0391}$ |

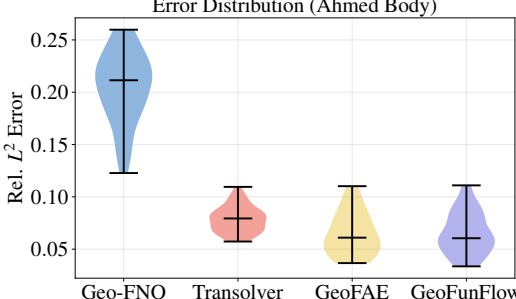

Figure 13: Violin plots of test errors for the Ahmed body dataset.

