# OpenReview forum: "GeoFunFlow: Geometric Function Flow Matching for Inverse Operator Learning over Complex Geometries"
_ICLR.cc/2026/Conference — Submitted to ICLR 2026_

### Official Review · Reviewer_bSce · 2025-10-25

**Soundness:** 2
**Presentation:** 2
**Contribution:** 2
**Rating:** 2
**Confidence:** 3

**Summary:**

The paper proposes a transformer-based autoencoder designed to reconstruct a continuous function from sparse point observations. The model employs Perceiver blocks to encode the observed samples into a latent representation, from which it queries different spatial positions to predict function values across the entire domain. In addition, the authors introduce a rectified field model in the latent space to endow the system with uncertainty estimation capabilities.

**Strengths:**

- Strong reconstruction performance: The proposed autoencoder demonstrates excellent reconstruction quality across multiple datasets, indicating that the model effectively captures the underlying structure of the target functions.
- Incorporation of uncertainty estimation: The paper makes a meaningful attempt to introduce uncertainty estimation by leveraging the stochastic nature of generative models based on flow matching.

**Weaknesses:**

- Unclear focus between autoencoder and generative components: The presentation of the method is somewhat confusing. While the paper frequently refers to flow-based prediction, the main contribution and experimental analysis appear to focus primarily on the autoencoder. The role and benefit of the generative (flow matching) component are only briefly demonstrated and not emphasized in the introduction.
- Uncertainty estimation reliability: In the appendix (Figures 7 and 10), the uncertainty maps show that the flow-matching model remains highly confident in regions where reconstruction errors are large, which suggests that the uncertainty estimation may not be reliable.
- Limited baseline comparison: Although the related work section (Section 2) cites several studies on continuous modeling of PDE solutions, the experimental comparison includes only two of them, along with mostly interpolated discretized baseline models. The rationale for excluding other relevant methods, e.g. those based on neural fields, is unclear. The authors should either better position their approach with respect to these omitted baselines or include a broader set of comparisons.
- Figure-text correspondence: The relationship between Figure 2 and the method description in Section 5 is unclear. The figure gives the impression that \(z_1\) corresponds to \(c\) (which should be \(z_c\)). It would help to visually separate the autoencoder from the rectified flow model to clarify the structure and data flow of the proposed method.

**Questions:**

- Improve the correspondence between Figure 2 and Section 5, as mentioned in the weaknesses. The visual diagram should clearly reflect the structure described in the text with same notations.
- Provide an explanation for why several relevant continuous-space models were excluded from the comparison or add some of the mentioned methods to the baselines.
- Expand the discussion on uncertainty estimation. It would be helpful to include a more detailed analysis of how well the estimated uncertainty correlates with reconstruction errors and whether the model captures epistemic or aleatoric uncertainty.

---

### Official Review · Reviewer_HsZe · 2025-10-29

**Soundness:** 2
**Presentation:** 3
**Contribution:** 2
**Rating:** 4
**Confidence:** 4

**Summary:**

GeoFunFlow proposes a new architecture to solve inverse problem. The method makes use of an auto-encoder based on a perceiver to encode irregularly sampled function onto a fixed latent code and uses a cViT based decoder. Finally, the method uses a flow based model that opers on the tokens to forecast the solution and provide uncertainty errors. The method is described and a theoretical analysis is provided as well as some evaluation on benchmarks.

**Strengths:**

- The paper is well detailed, clear and easy to follow.
- All experimental details settings are proposed, making the experiment reproductible.
- A theoretical analysis is proposed to strengthen the method.

**Weaknesses:**

- In my opinion, the main weakness to the paper, lies in its originality. The architecture is a combination of several existing modules. Moreover, similar architectures have been proposed. In particular, AROMA that is already mentioned by authors, that already uses Perceiver + Diffusion for handling irregular grid and forecasting PDEs. In my understanding, the main difference between these 2 models lies in the decoder: GeoFunFlow uses a cViT whereas AROMA uses INRs. The method is not compared to this baseline neither in terms of architecture design nor in terms of performances.
- Experiments: The experimental setup considers 6 baseline models. Out of these 6 models,  only 2 are designed to handle irregular grids. Others make use of interpolation. This weakens the results as a lot of models have been proposed to manage irregular geometries, (as detailed in the related work section): CORAL, AROMA, GINO, PCNO (already cited in paper), RIGNO [1], UPT [2] and many other more recent work. Moreover, I think more recent baselines considering Diffusion/flow matching could be considered such as DiffusionPDE already mentioned in the manuscript (by removing the PDE loss?) or [3-4].

[1] RIGNO: A Graph-based framework for robust and accurate operator learning for PDEs on arbitrary domains, Sepehr Mousavi, Shizheng Wen, Levi Lingsch, Maximilian Herde, Bogdan Raonić, Siddhartha Mishra, 2025.

[2] Universal Physics Transformers: A Framework For Efficiently Scaling Neural Operators, Benedikt Alkin, Andreas Fürst, Simon Schmid, Lukas Gruber, Markus Holzleitner, Johannes Brandstetter, 2024

[3] Scalable diffusion models with transformers, William Peebles and Saining Xie, 2022.

[4] Generative latent neural pde solver using flow
matching, Zijie Li, Anthony Zhou, and Amir Barati Farimani, 2025.

**Questions:**

-	Could you explain the differences with respect to AROMA ? This baselines seems very close to the proposed method in my opinion, except for the decoder architecture.
- How does GeoFAE/GeoFunFlow perform in comparison with other irregular-geometries-designed baseline? With respect to other diffusion baselines?
-	Could you provide run time/inference time comparison ? Since the model makes use of a Diffusion process, that involved an iterative process, I think a comparison on this aspect is important.
-	How does the method scales in terms of memory consumption ? Perceiver, and more generally attention bases technique have bad scaling behavior with respect to the trajectory length/number of points?
- How are the latent queries size are chosen?
- Unfortunately, I am not an expert in the proposed theoretical analysis that makes use of tools i am not well-aware of. However, line 358, it is stated that we can analytically compare a convergence rate. Is it possible to add the theoretical rate to the figure? This will help for clarity. Is this convergence rate really verified in practice on quasi-uniform grids (as stated lines 352-359) ?
- Regarding the note on benchmark (line 967-971), Why couldn't GeoFunFlow be conditioned only on the geometry of the PDE? By training the model to regress static PDE data, the model could encode itself the dynamic and thus be able to reconstruct the solution based only of the geometry?
- Regarding training time provided in appendices: Why is the training time of GeoFAE sometime higher than GeoFunFlow? In my understanding, GeoFunFlow include GeoFAE, thus include also the training time of GeoFAE?

### Minor comments
- Could you precise, what '-' are meaning in Table 1?

---

### Official Review · Reviewer_e4ca · 2025-10-30

**Soundness:** 2
**Presentation:** 3
**Contribution:** 2
**Rating:** 4
**Confidence:** 4

**Summary:**

This paper addresses inverse problems governed by PDEs on complex geometries, where the goal is to reconstruct physical fields from sparse and noisy sensor measurements. The authors propose GeoFunFlow, which combines a geometric function autoencoder (GeoFAE) with a latent diffusion model. The GeoFAE uses a Perceiver module to encode variable-sized meshes into fixed-dimensional latent representations and a CViT-inspired decoder for continuous field reconstruction. A Diffusion Transformer then performs posterior sampling in the latent space using rectified flow. The method is evaluated on five benchmarks (Darcy, Cylinder, Plasticity, Airfoil, Ahmed Body) and compared against both regular-grid baselines and geometry-aware methods.

**Strengths:**

- The architecture is sound, and inspired from prior works that also use diffusion ([1], [2])
- The methods (both GeoFAE and GeoFunFlow) obtains good results on the different datasets.
- The theoretical result with $m$ quasi-uniform sensors is interesting, as it clearly shows that the sampling density influences the Wasserstein distance.

[1] Serrano et al., AROMA: Preserving Spatial Structure for Latent PDE Modeling with Local Neural Fields, 2024.
[2] Zhou et al., Text2PDE: Latent Diffusion Models for Accessible Physics Simulation, 2024.

**Weaknesses:**

- The architecture is positioned as novel, but most blocks are taken from other papers as mentioned by the authors, and the methodology, though at first glance is framed as an inverse problem is actually, solving a conditional forward problem. This greatly diminishes the impact of the paper, as the training is simply a two-stage process of a diffusion model conditioned by some observations.
- The GeoFunFlow variant does not really help and is on most datasets comparable and even inferior to the simple auto-encoder, this is not a very compelling evidence for the framework, given that the main novelty of the work is put on the capability of solving "inverse" problems.
- A minor detail about the methodology and concerning the term "rectified flow". It appears to me that, what is described in the paper is actually flow matching, though these concepts are close to each other. I believe that in rectified flow, we rectify the trajectory by sampling non-random pairs of noise and and observations.
- Most baselines are for regular grids (DPS, ViT, Unit, FNO), and they should not be included. More relevant baselines, such as UPT, AROMA or Text2PDE (Zhou et al, 2024, their auto-encoder) should be used instead.

**Questions:**

1. What is the motivation for adding the flow-matching component if it does not improve reconstruction accuracy?
2. If its main role is uncertainty estimation, why focus the evaluation on deterministic reconstruction metrics ?
3. Could you provide quantitative calibration results for uncertainty and compare with baselines such as ensembles or MC dropout?
4. Could you report results using averages over multiple diffusion samples (e.g., 10–20)?
5. How does your method differ technically from AROMA which has a very similar architecture and also trains a diffusion model in the latent space?

---

### Official Review · Reviewer_Wrbm · 2025-11-01

**Soundness:** 3
**Presentation:** 3
**Contribution:** 3
**Rating:** 6
**Confidence:** 4

**Summary:**

This paper proposes GeoFunFlow, a geometric functional autoencoder combines with a latent diffusion model for solving inverse problems on complex geometries. The method combines a  Perceiver module in encoder for varying discretizations, a CViT in decoder for continuous evaluation, a conditional DiT for posterior approximation. Experiments across five standard benchmarks show SOTA reconstruction accuray compared to strong baselines such as DPS, ViT, UNet, FNO, Geo-FNO and Transolver. The framework can also provide uncertainty estimates.

**Strengths:**

1. Novel geometric functional autoencoder for PDEs inverse problem on complex geometries.
2. A latent diffusion model trained via rectified flow for uncertainty quantification.
3. Comprehensive experiments and ablation studys verify the effectiveness of the method proposed.
4. SOTA accuracy on five benchmarks.

**Weaknesses:**

1. There is explanation on why GeoFAE has better acurracy than GeoFunFlow, but there is no explanation on why GeoFunFlow has a better performance than GeoFAE on Ahmed Body.
2. Lack of details on uncertainty quantification, in Figured 7, 9, 10, there are still many visual differences between GeoFunFlow:Error and GeoFunFlow:Std (uncertainty). More details need to be delivered to explain how uncertainty quantification reflect the  magnitude and spatial distribution of errors.
3. The paper only compares with DPS and ViT, lacking experimental analysis against more recent diffusion-based PDE methods.

**Questions:**

1. Why GeoFunFlow's performance improves after adding the diffusion module? Does this happen frequently in the Ahmed Body dataset, or only occasionally? Are there cases in other datasets where GeoFunFlow outperforms GeoFAE?
2. Can you provide more explanations on uncertain quantification and effective demonstrations of uncertainty?

---

### Meta-Review · Area_Chair_5CFW · 2025-12-30

**Summary:**

This paper proposes a flow-matching based technique for solving inverse problems arising from PDEs. Reviewers generally appreciate the empirical results, clarity of presentation, and theoretical results.

Reviewers raise the following weaknesses:
- [W1] Limited comparison against relevant baselines (WRbm, e4ca, HsZe, bSce)
- [W2] Insufficient exploration of the uncertainty quantification component (WRbm, e4ca, bSce)
- [W3] Concerns about novelty (e4ca, HsZe)

Overall, the key concerns regarding limited baseline comparisons, insufficient analysis of the uncertainty quantification component, and unclear novelty remain unaddressed, particularly given the absence of a rebuttal. Based on these unresolved issues, I recommend that the authors carefully consider the feedback provided by the reviewers and resubmit their article at a later date.

**Reviewer Concerns:**

The authors did not post a rebuttal, and hence the raised weaknesses still remain.

**Reviewer Scores:**

The reviewers are unlikely to have raised their scores given that no rebuttal was posted.

---

### Decision · Program_Chairs · 2026-01-26

Reject